# Analysis of Injury Patterns in Men’s Football between the English League and the Spanish League

**DOI:** 10.3390/ijerph191811296

**Published:** 2022-09-08

**Authors:** Juan Carlos Argibay-González, Christopher Vázquez-Estévez, Alfonso Gutiérrez-Santiago, Adrián Paramés-González, Xoana Reguera-López-de-la-Osa, Iván Prieto-Lage

**Affiliations:** 1Observational Research Group, Faculty of Education and Sport, University of Vigo, 36005 Pontevedra, Spain; 2Education, Physical Activity and Health Research Group (Gies10-DE3), Galicia Sur Health Research, Institute (IIS Galicia Sur), SERGAS-UVIGO, 36208 Vigo, Spain

**Keywords:** injury, football, pattern, video analysis

## Abstract

Background: Injuries in professional football lead to reduced team performance and large financial losses. The aim of this study was to analyse injuries in the two best team competitions in the world (LaLiga and Premier League), establishing similarities and differences, as well as determining injury causation patterns. Methods: A total of 277 on-field injuries requiring a substitution were analysed (142 in the Spanish league and 135 in the English league). The analysis was performed using traditional statistical tests (frequency analysis, chi-square test) with SPSS 25 and a T-Patterns sequence analysis with THEME 5.0. Results: In the Spanish league, there were a similar number of injuries in the first part of the season as in the second part of the season, while in the English league, they are more frequent in the first part of the season. In the Spanish league, injuries are more frequent in the first half of the match, while in the English league, they are more frequent in the second half. The type of player most frequently injured was the defender. Most of the injuries occurred without the presence of an opponent. The accumulated minutes during the season affect injuries of the muscular type. Conclusions: The most common type of injury in both leagues was a strain, followed by a sprain and a contusion. Although common injury patterns can be established between the two leagues, there are notable differences. One of the factors is determined by the English league calendar (many matches at Christmas). In Spain, there were more muscular injuries that were not caused by the opponent, while in England there were more tackling injuries. Age is a risk factor in the Spanish league. In this league, there is a moderately significant relationship between the number of injuries and the points won.

## 1. Introduction

Professional football is a contact sport with an associated risk of injury [1,2]. Injuries are the main factor affecting a player’s availability [3], causing a problem in the training–competition process, with consequences on their fitness and performance [4]. Research showed that a professional football team can expect around 50 injuries per season, resulting in approximately two injuries per player [5]. The incidence of match injuries was almost 10 times higher than the incidence rate of training injuries. The injury incidence rate in the five major European professional leagues (Spain, England, Italy, Germany and France) was not different from that of professional leagues in other countries (6.8 vs. 7.6 injuries/1000 h of exposure, respectively) [6]. The study of injuries is one of the major concerns in the scientific field of football, as evidenced by the numerous published studies [6].

The Premier League (England) and LaLiga (Spain) are the two most important football competitions in Europe [7]. Injuries have a direct relationship with the length of time a player is absent and the economic impact it has on the club. Thus, a study in the English Premier League has determined that teams will pay more than GBP 100,000,000 in wages for players who will be unavailable due to an injury for more than 30 days [8].

Concerning injury studies in football, one of the main problems is that there is no uniform injury reporting system, even in the major European leagues, which makes it difficult to analyse [9]. In addition, researchers use different ranges and classifications when categorising injuries, rather than following the UEFA model [10]. Another problem is that most of the research carried out is descriptive, without determining the mechanism by which the injury occurs.

The scientific literature approaches the study of injuries in terms of both intrinsic and extrinsic risk factors [11]. Intrinsic factors include the biological characteristics of the players. Age is a variable that has been studied, but the ranges used vary according to the research. Despite the fact that their study is not uniform, there are studies that indicate that, as the age of a footballer increases, the probability of suffering an injury will also raise each year [5,12]. It has also been observed that the medical history of the football player with previous injuries in the same body location will increase the incidence of injury [11,13]. At the same time, a higher risk of injury on the dominant side of the player has been found [5,14,15]. Regarding the location of the injuries, studies show different classifications, however, most researches focus on the lower limbs, since these are the extremities that present the highest percentage of injuries [15,16,17,18]. The anatomical region where most injuries occur is the thigh, followed by the knee and ankle [6]. Previous epidemiological studies have reported that hamstring muscles are the muscle group most frequently injured in professional football players [16,19].

On the other hand, extrinsic factors are those related to the context or the time spent practising sport. Regarding the training–competition process, the incidence of injuries is higher during competition [3,16,19,20,21,22,23,24]. In this section, it is important to note two moments of the season that will have an influence on the injury rate of footballers. A common peak of incidence in England and Spain is the pre-season, a period of the season in which there is a greater volume of training, in order to arrive in the best possible condition for the start of the competitive period. The incidence of injuries during training is higher during July and August, compared to the training phases of the rest of the season [21]. In England, there is another peak of incidence in December, a period in which a greater number of matches are concentrated [18]. Regarding the timing of the match, several studies show a higher incidence of injuries during the second half, compared to the first half [15,21]. In addition, it is observed that there is a higher risk of injury towards the end of each half of the match [15,16]. The position of the players also influences the frequency of injury. Different studies show similar percentages, being higher in midfielders, followed by defenders, forwards and goalkeepers [14,22,25,26].

The importance of the present study lies in the fact that it provides innovative knowledge about injury patterns, also taking into account causal factors. This knowledge, as some authors point out, is the first step towards establishing an injury prevention programme [27]. Therefore, we believe that this study will be useful for football clubs in helping them to develop such programmes, as it has been shown that they can reduce injuries by up to 39% [28]. Furthermore, this reduction will cut down the aforementioned economic impact. Based on this, the object of study of the present research is to analyse and determine injury patterns that occurred in LaLiga (Spain) and the Premier League (England) during the 2018/2019 season and to look for differences between these two competitions.

## 2. Materials and Methods

### 2.1. Design

Observational methodology [29] was used to detect patterns in football players’ injuries. The observational design [30] used was nomothetic (all injuries were studied independently), follow-up (one season) and unidimensional (one level of response). A series of decisions about the participants, instruments and the analytical process is derived from this design.

### 2.2. Participants

The participants of this study were the players of the first division of the Spanish and English football league for men in the 2018–2019 season who suffered an injury for which they were withdrawn from the field. A sample of 277 (142 and 135, respectively) injuries was obtained in the 38 league games. The information about the injuries was taken from the official medical records of the clubs and from TransferMarkt, a valid sports database [1] that provides information about the injuries. The injuries were analysed in accordance with the ethical principles of the Helsinki Declaration using audio-visual material of public domain [31]. According to American Psychological Association [32], an observational study in a natural environment, with public videos obtained from the mass media that does not imply experimentation, does not require an informed consent from the competitors. The study was approved by the Ethics Committee of the Faculty of Education and Sport Science (University of Vigo, Application 02/1019).

### 2.3. Instruments

The observation instrument used for this study is an adaptation of the OI-INJURIES-FOOTBALL [33], a category system that contemplates a collectiveness of criteria that will allow us to the determine the football injuries characteristics [30]. Each dimension gives rise to a system of categories that accomplish the conditions of exhaustiveness and mutual exclusivity (E/ME). A detailed description of the observation instrument appears in Table 1, where the criteria, categories and subcategories of the instrument are shown. In Figure 1, the established field areas are detailed.

The categorization of the type of injury followed the UEFA model [10]. To designate the part of the body where the injury was produced, a classification endorsed by other authors was used [34]. Four demarcations were used for the analysis [1] and divided the match into 15 min intervals [35]. The cause of injury followed the model established by other researchers [36].

The recording instrument used was the LINCE v1.4. (Lleida, Spain) [37]. This software has been used in numerous investigations about football [38,39].

### 2.4. Procedure

The sample was obtained through the platform Wyscout [40], which is a scouting platform for football.

After an adequate training in the use of the register instrument and the observational instrument, the register of the data was carried out by two observational experts. To guarantee the rigor in the codification process [41], the quality of the registered data was controlled through a calculation of the intra- and inter-observers compliance using the Cohen’s Kappa coefficient calculated through the software LINCE.

Intra-observer agreement was performed on the total number of lesions (*n* = 277), obtaining a kappa value of 0.97 for observer 1 and 0.95 for observer 2. Subsequently, inter-observer agreement reached a kappa value of 0.93. The data used were those of observer 1.

After registering all the injuries, an Excel file is obtained with the sequence of all the codes of the registered behaviours, with its temporality and duration expressed in frames. The versatility of this file allowed us to conduct successive transformation for the different analysis [33].

### 2.5. Data Analysis

All the statistical analysis was carried out using the IBM-Statistical Package for the Social Sciences, version 25.0 (IBM-SPSS Inc., Chicago, IL, USA). It calculated the relationship between the distinct categories that were studied by using the test of the Chi-square (χ2). Pearson’s correlation coefficient was also calculated to analyse the relationship between the club’s league ranking points and the number of injuries (0.90 to 1.00 very high; 0.70 to 0.89 high; 0.50 to 0.69 moderate; 0.30 to 0.49 low; 0.00 to 0.29 negligible) [42]. The statistical significance was assumed as *p* < 0.05.

To analyse the patterns of injury, the detection of T-Patterns (temporal patterns) was carried out by the Theme v.5.0 [43] with the significance level of 0.005, which means that the percentage of accepting a critical range due to chance is 0.5%. The minimum number of occurrences in the search of T-Patterns was five (the minimum possible without a mistake in processing the information because of an excessive number of series). Furthermore, the reduction in redundancies has been activated to 90% to avoid the occurrence of similar T-Patterns. A T-Pattern is essentially a combination of events that occur in the same order, with the consecutive time intervals between consecutive pattern components remaining relatively invariant, it being assumed, as a null hypothesis, that each component is independently and randomly distributed over time. In the results tables, they are presented in mathematical format with their occurrence (o) and length (l) (for more information, see the reference manual [44]). This software reveals hidden structures and non-observable aspects in sports science [45].

## 3. Results

### 3.1. Statistical Analysis

In Table 1, a descriptive analysis and χ^2^ test intra-criteria are presented.

In LaLiga, there was a similar number of injuries between matchdays 1–19 and matchdays 20–38 (52.1–47.9%), with April being the month with the highest incidence (16.9%). In the Premier League. there was a higher number of injuries in the first part of the season (74.8–25.2%), especially in the month of December (29.6%).

In the Spanish league, there were more injuries in the first half (57.0–43.0%), while in the English league the opposite happened (39.3–60.7%), although significant differences are only found in the English league. In both leagues, there were more injuries in the defensive zone, but with very similar values, compared to the offensive zone, with no significant differences in both competitions. In both leagues, it was found that the age group 26–30 years (47.9% and 51.9%) and the position of defender (53.5% and 38.5%) reported more injuries than the other groups. It was also evident in both leagues that the body parts where players are more frequently injured are the thigh (44.4–35.6%), the knee (13.4–13.3%) and the ankle (10.6–17.8%). There was no coincidence in the leg where the injury occurred, being the left leg (46.5%) in the Spanish league and the right leg (45.2%) in the English league. In both countries, it was more frequent that the injury occurred without the presence of an opponent, although with different values (69.7–57.0%). In the Spanish league there were more injuries as the accumulated minutes of play increased during the season. This was not the case in the English league, where there were no significant differences in the variable cumulative minutes of play during the season. The most common injuries in both leagues were strains (50.0–28.9%), sprains (19.7–20.7%), contusions (9.2–20.0%) and overuse (9.9–17.0%), although with different percentages.

Statistically significant differences were observed between LaLiga and Premier League in the following criteria: type of injury, moment in the season, month, time, age, position, causer, total accumulated minutes and accumulated minutes after resting. Likewise, no differences were found in the criteria stadium, zone, injury location, injured leg, player laterality and how the injury is caused.

Table 2 shows the frequency and percentage of players who have participated in the league competition in their respective country (only those who have played at least 1000 min have been taken into account to avoid bias, since too many footballers had played very few minutes), as well as the percentage of injured players in the study. The data were stratified by age (it does not include goalkeepers) and position, showing also the difference between the two percentages in both cases.

Table 3 presents, for each league, an analysis between the points won by each club and the number of total injuries at the end of the season.

No statistically significant relationship was found between points won in the ranking and number of injuries in the English league (r = 0.442; *p* = 0.182). In contrast, a statistically significant relationship was found in the Spanish league (r = 0.534; *p* = 0.015), although it was moderate.

### 3.2. Identification of Temporal Patterns (T-Patterns)

#### 3.2.1. General Description of T-Patterns

Table 4 shows an analysis of the number of patterns found through a selective search with five occurrences. As a selection criterion, the presence of the position category was used (defence, midfield and forward), combined with one of three most frequent injuries (strain, sprain, contusion or overuse).

Table 5 included below shows the most relevant T-patterns organised by league, position and type of injury.

#### 3.2.2. T-Patterns in Defenders

In both leagues, the most common injury is a hamstring rupture and it occurs in the absence of an opponent. In La Liga, strain injuries are more frequent in the first half (55%), while in the Premier League they are more frequent in the second half (56%). In Spain, the minutes accumulated during the season are relevant, while in England, it is the time accumulated after resting. Age was a key factor, since in Spain up to 55% of injuries of this type occurred in players over 30 years of age (69% in England in players between 26 and 30 years of age).

Regarding sprains, it is observed that they were more frequent in the first part of the season (82–77%, respectively), in the knee (55–69%) and caused by the opponent (73–77%). In the Spanish league, up to five out of the eleven sprains were a rupture of the cruciate ligament.

Contusions were mostly caused by the opponent in both leagues, being more common in the second part of the season in the Spanish league and in the first part of the season in the English league.

The overloads in Spain occurred mainly in the second half (89%) and in the hamstring area (67%) of the thigh (78%) of the left leg (67%). There was no relevant data for England.

#### 3.2.3. T-Patterns in Midfields

In relation to strains, in the Spanish league, they were more frequent in the first part of the season (61%), in players over 30 years of age (44%), in the first half (67%), without contact with the opponent (89%), after a sprint (50%). In the English league, they were more common in the second half of the season (63%), in the second half (75%), in players between 26 and 30 years of age (63%), without contact with an opponent (63%) and in the left leg (75%). Up to 75% of the strains were hamstring fibrillar ruptures.

In both LaLiga and the Premier league, sprains were most common in the first part of the season (86%), caused by the opponent (71%), in the ankle area (57–86%). In England they occurred mostly in the right foot (86%) and between 0–200 min accumulated after a rest period (86%).

There were no patterns with an occurrence of five or more in this position in relation to contusions and overloads in Spain. Contusions in England occurred in the second half (90%), on the lower extremity (80%) and caused by the opponent (100%). The overloads occurred in the first half of the season (69%), in the second half of the match (77%), in the calf area (54%) and after a period of 0–200 min accumulated after resting (62%). Up to 54% of the overloads occurred in players aged 21–25.

#### 3.2.4. T-Patterns in Forwards

The most frequent strain in both Spain and England was a hamstring fibrillar rupture (69–86%). The month in which most of these injuries occurred was October for the first case and December for the second one (38–36%), being more frequent in the first part of the season in both leagues (62–57%). They occurred more regularly in the first half in Spain and in the second half in England (77–57%). Likewise, in the Spanish league they occurred after an accumulated period of between 0–200 min after a rest period, while in the English league they occurred after accumulating between 1000–1500 min during the season.

In both the Spanish and English leagues, ankle sprains were frequent (60–75%), although Spanish forwards were injured without contact with the opponent (80%) and English forwards after contact with the opponent (63%). In both cases, they were more common in the second half of the match (70–88%).

There were no patterns with an occurrence of five or more in the forward position for contusions and overloads in Spain. In England, it was evident that the contusions were caused by an opponent and in the offensive zone in 100% of cases. Overloads were more common in the first part of the season (83%), in the thigh area (83%) and without the presence of an opponent (100%).

## 4. Discussion

The purpose of this research was to determine the injury pattern of the football player in the Spanish and English men’s league.

The most frequent injury observed in both LaLiga and the Premier League was the strain (50.0–28.9%), specifically the hamstring fibrillar rupture (29.6–23.7%). The percentage of the type of injury is very different between the two leagues, although there is coincidence in the subtype of injury. Most of the authors agree that strains are the most frequent injuries, giving values between 40% in the Spanish league and 35% in the English league [17,46]. In terms of injury incidence, approximately six players out of each squad will injure a hamstring each season in professional soccer [47]. Intrinsic factors causing this injury (such as age and medical history) are risk factors [48], although they are not modifiable. Modifiable risk factors include hamstring weakness, fatigue and lack of flexibility [49], with the strength imbalance between the hamstrings (eccentric) and the quadriceps (concentric) being the most supported by the literature. [50].

In previous investigations into the Spanish league [33,35], it was observed that there were more injuries in the first part of the season (approximately 55%). The same occurred in the season of this analysis and in both leagues (52% in Spain), especially in the English league, where the injury rate was close to 75%. In the English league, there was a clear increase in injuries in the month of December (29.6%), something that may have occurred due to the increased number of matches at Christmas (up to nine matches in the English league, compared to a maximum of six in the Spanish league). In the Spanish league, although December was also one of the months presenting more injures, the percentage was not as high as in England (13.4%), probably due to the fact that there were no matches between the 24th and the 30th of December because of holidays. A study consulted in which a prospective analysis was carried out over 14 years showed that the concentration of matches in a short period of time is not the variable that affects the total number of injuries, but rather the lack of player rotations that allow a rest of more than six days between matches [51]. This could explain the increase in injuries in that part of the season. In Spain, October was a month with many injuries (13.4%), evidence that has already been observed in other studies [25,52]. April was the month with a higher percentage of injuries (16.9%), a number, which is also reported in the scientific literature on this league [33,35]. Once again, the accumulation of matches, the absence of player rotations and competitive stress (uncompleted objectives at that critical moment of the season), as well as the change in weather conditions (more heat and dryness) could be the causal factors. [53,54].

The relationship between the injury and the time of the match in which it occurred was very different in the Spanish and English leagues. In Spain, in all quarters of play there was a stable injury rate, except in the last two quarters of play, where it dropped. In England, the opposite happened, with the last 30 min of the match reporting the most injuries. Longitudinal research conducted on the top 14 European teams during the 2011–2018 seasons [16] shows that injuries tend to occur towards the end of matches. However, specific studies related to the Spanish league show a low injury rate in the final period of the match in this country [33,35]. The explanation to this may be related to factors, such as physical preparation, preventive training, intensity of play, style of play, condition of the fields and substitutions made, even though all these indicators have not been studied so far [6]. End-of-match injuries could be minimised by the new rule change where up to five substitutions are allowed, although this is an aspect to be studied in future research.

The specialised literature [33,35] has reported a higher number of injuries in the defensive zone of the field (59–68%), something that has also happened in our case in both leagues (54.2% and 51.1%), although without substantial differences, with respect to the offensive zone. This could be because the physical demands are different depending on the position or the specific task required by the player at each moment of the match [55]. For example, defensive actions require more tackling or jumping headfirst [56] and also a greater number of high-intensity actions with changes of rhythm to avoid dangerous situations for the attackers.

Age is a relevant risk factor, thus, the number of injuries increased in the 21–25 and 26–30 age ranges and then decreased in the over-30 age group (due to a lesser number of players in this age range). Previous research has shown that increasing age leads to a higher number of injuries, especially muscular injuries [12,16,35,52]. The analysis in Table 2 corroborates the published scientific evidence. In any case, this circumstance has been found to be clearer in the Spanish league than in the English league, where this variable has not been so decisive. In Spain, the relative injury incidence rate increases notably from the age of 25 onwards (+5.0% in the 26–30 age group and +9.3 in the 30+ age group), but in England, the increase was less significant (+2.6% and +0.6%, respectively). It would be important to take this information into account in injury prevention programs, which should be more specific as age increases. In addition to this, it is important to control the minutes accumulated by players throughout the season, and, especially, the rest days between competitive matches, again with special mention to players over 25 years of age.

Studies on football and injuries do not agree on which position has the highest injury rate. For example, in the Italian league [52] it is indicated that any field position (excluding goalkeepers) is equally affected. In the French league, forwards [57], and in the German league, midfielders are the most frequently affected [1]. In Spain, the studies carried out indicate that defenders are the players most affected by injuries [33,35], which is consistent with the findings of this research in both LaLiga and the Premier League, although in Spain the percentage is higher (53.6% vs. 38.5%). Based on the data from previous research and the analysis in Table 2, we cannot affirm that there is a field position that requires specific injury prevention treatment. It is noticed, in relative terms, that goalkeepers are the players who suffer less injuries (−7.9 and −6.5). In the Spanish league, the relative incidence rate is higher for defenders (+16.8), although this does not occur in the English league (+0.1), where forwards are the players with the highest incidence (+5.0). In the Spanish league the relative ratio in this position was negative (−4.9). Midfielders in Spain had a lower ratio than they should have (−4.0), while in England it was higher (+1.4). Probably, although it cannot be confirmed, age or lack of rest between matches are the most important risk factors, even if this requires further study. It may also be due to the idiosyncrasies of each country’s style of play [58].

After analyzing the distribution of injuries according to the ranking of the clubs (Table 3), we can point out that there is no significant relationship between the number of injuries and the points won in the rankings in the English league. In the Spanish league, there is a moderately significant relationship.

There was a clear tendency for injuries to occur on the lower extremity, a fact already reported in the literature in the past [16,21], with thigh injuries being more common (44.4% in Spain and 35.6% in England), which was also contemplated in previous research [21,52,59]. Muscle injuries are more frequent, with hamstring fibrillar rupture being the most common, an aspect that has been confirmed in numerous studies. Imbalance between the thigh musculature, age and medical history are the main risk factors for this injury [12]. The importance of this type of injury has been specifically studied. Fortunately, several authors have established different mechanisms for its prevention [50].

In both leagues, most injuries occur without the presence of an opponent, occurring after a sprint. T-Patterns corroborate that injuries after a sprint are mostly fibrillar ruptures, which would explain this high percentage, being the most common subtype of injury. Previous studies confirm this association and even point to an annual increase of 4% in recent years of this type of injury [60]. It is also observed that in England, there are more injuries due to tackling (19.9% vs. 30.4%). Recent research found that the English league is the most aggressive of the five major European leagues, which may explain the data previously reported [61].

In a recent study of the Spanish league [35], researchers observed that injuries increased as players accumulated more minutes in the season, a trend that is confirmed by our results. In contrast, in the English league, there were numerous injuries without players accumulating many minutes. This may be due to the fact that in this early part of the season there is a high concentration of matches during the Christmas period.

If we refer to the injuries that occurred according to the minutes accumulated after a rest of more than 7 days (at competition level), we found that, as indicated in previous studies [33,35], most of the injuries occurred between 0–200 min accumulated after this rest. Coaches usually leave out the most overloaded players when they are at their muscular limit, and then they come back to play in the next match when they are not yet in full physical condition [62]. This could possibly be the motive.

### 4.1. Limitations and Future Perspectives

This study only considers injuries that occur during league matches that require player substitution (although it does take into account the accumulated minutes of play in all competitions). It would be interesting to extend the study to all team competitions and even those injuries that occur during training. The analysis could also be extended to other competitions in other countries. Future research should approach the analysis of injuries in football with longitudinal studies.

### 4.2. Practical Implications

Due to the high incidence of injuries in older players, especially muscular injuries, specific interventions are necessary to minimise the risk while maintaining a highly competitive level. It is recommended to monitor the imbalance between the quadriceps and hamstring musculature.

The competitive calendar is highly concentrated, but it has been shown that rotations are necessary to avoid injuries. Coaches should be aware that playing two games in less than 7 days increases the risk of injury, especially in older players. It would be important that squads are as balanced as possible, with at least two players per position and that there is a distribution of minutes between them throughout the season as far as possible.

Injury prevention programmes are important because they help to reduce the risk of injury. They should be included in training sessions for all positions and ages.

## 5. Conclusions

Although there were similarities between the Spanish and English leagues in terms of injuries, there were also clear differences.

The most common type of injury in both leagues was strain, followed by sprain and contusion. There were also a high number of overuse injuries, which were more frequent in the Premier League. In Spain, there were more muscle injuries (including strains and overuses) than in England. In both leagues, the most frequent injury was hamstring rupture, so specific work to prevent this type of injury is recommended. In the English league, a higher number of contusions were observed. The number of sprains was similar, with knee sprains being more common. Tackling injuries were more common in England, while non-contact injuries after a sprint were recorded more often in Spain.

The calendar affected the number of injuries. The Christmas period saw the highest number of injuries in England due to the high number of matches scheduled. The Spanish league had a balanced number of injuries throughout the season (especially in April, where several competitions were also concentrated). It was also found that the defender is the player with the most injuries, being more accentuated in Spain. The accumulation of minutes throughout the season was clearly linked to injury in the Spanish league but not in the English league. After a rest of at least 7 days without competing, the players who got injured did so after playing between 0–200 min in the majority of cases. This circumstance requires further study, but it is an aspect of great relevance.

In the Spanish league, there was a higher risk of injury as the age of the player increased. In the English league, age was not such a determining factor. The position with the highest risk of injury in Spain was defender, while in England it was forward. There is no significant relationship between the points won and the number of injuries in the English league, and a moderate one in the Spanish league.

## Figures and Tables

**Figure 1 ijerph-19-11296-f001:**
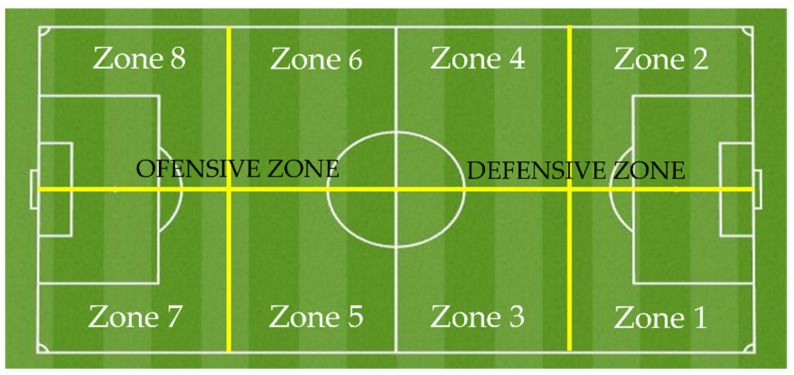
Field zones.

**Table 1 ijerph-19-11296-t001:** Descriptive obtained values and the link between the categories.

Criteria	Category	Subcategory	Spain		Intra-Criteria	England		Intra-Criteria	Inter-Criteria
			*n*	%	χ^2^ (*p*)	*n*	%	χ^2^ (*p*)	χ^2^ (*p*); Contin. Coef.
**Location and type of injuries**									
Injury location	Head, face or neck		6	4.2	182.634 (0.000)	10	7.4	80.881 (0.000)	13.075 (0.109); 0.212
	Upper limb		6	4.2		9	6.7		
	Trunk-back		1	0.7		5	3.7		
	Lumbopelvis		18	12.7		8	5.9		
	Thigh		63	44.4		48	35.6		
	Knee		19	13.4		18	13.3		
	Calf		13	9.2		13	9.6		
	Ankle		15	10.6		24	17.8		
	Foot		1	0.7		10	7.4		
Injured leg	Right leg injuries		59	41.5	0.392 (0.531) ^a^	61	45.2	1.309 (0.253) ^a^	1.595 (0.207); 0.082 ^a^
	Left leg injuries		66	46.5		49	36.3		
	No leg injury		17	12.0		25	18.5		
Type of injury	SPRAIN		28	19.7	70.508 (0.000) ^b^	28	20.7	4.812 (0.186) ^b^	16.087 (0.001); 0.249 ^b^
	Ankle sprain		5	3.5		9	6.7		
	Knee sprain		13	9.2		16	11.9		
	Acromioclavicular sprain		2	1.4		-	-		
	Cruciate ligament rupture		8	5.6		3	2.2		
	STRAIN		71	50.0		39	28.9		
	Hamstring fibrillar rupture		42	29.6		32	23.7		
	Quadricep fibrillar rupture		9	6.3		-	-		
	Soleus-gastrocnemius fibrillar rupture		9	6.3		-	-		
	Adductor fibrillar rupture		11	7.7		7	5.2		
	CONTUSION		13	9.2		27	20.0		
	Head, face or neck contusion		2	1.4		3	2.2		
	Upper extremity contusion		-	-		3	2.2		
	Trunk-back contusion		1	0.7		5	3.7		
	Lower extremity contusion		10	7.0		16	11.9		
	FRACTURE		5	3.5		4	3.0		
	Head, face or neck fracture		2	1.4		2	1.5		
	Upper extremity fracture		2	1.4		1	0.7		
	Lower extremity fracture		1	0.7		1	0.7		
	DISLOCATION		2	1.4		5	3.7		
	Upper extremity dislocation		2	1.4		5	3.7		
	OVERUSE		14	9.9		23	17.0		
	Hamstring overuse		7	4.9		12	8.9		
	Adductor overuse		3	2.1		-	-		
	Quadriceps overuse		2	1.4		2	1.5		
	Gastrocnemius overuse		2	1.4		9	6.7		
	CONCUSSION		1	0.7		5	3.7		
	OTHERS (wound, etc.)		8	5.6		4	3.0		
	Not categorized injuries		7	4.9		-	-		
	Meniscus Tear		1	0.7		2	1.5		
	Achilles tendon rupture		-	-		2	1.5		
**Intrinsic injury risk factors**									
Age	<21 years		1	0.7	17.149 (0.000) ^a^	2	1.5	25.549 (0.000) ^a^	9.038 (0.011); 0.179 ^a^
	21–25 years		28	19.7		40	29.6		
	26–30 years		68	47.9		70	51.9		
	>30 years		45	31.7		23	17.0		
Position	Goalkeeper		2	1.4	27.829 (0.000) ^a^	1	0.7	1.806 (0.405) ^a^	6.729 (0.035); 0.155 ^a^
	Defender		76	53.5		52	38.5		
	Midfielder		30	21.1		41	30.4		
	Forward		34	23.9		41	30.4		
Player laterality	Right footed		100	70.4	28.971 (0.000) ^a^	95	70.4	33.800 (0.000) ^a^	0.311 (0.577); 0.034 ^a^
	Left footed		37	26.1		30	22.2		
	Ambidextrous		5	3.5		10	7.4		
**Extrinsic injury risk factors**									
Moment in the season	Days 1–19		74	52.1	0.254 (0.615)	101	74.8	33.252 (0.000)	15.332 (0.000); 0.229
	Days 20–38		68	47.9		34	25.2		
Month	August		4	2.8	18.704 (0.028)	18	13.3	82.111 (0.000)	37.047 (0.000); 0.343
	September		13	9.2		12	8.9		
	October		19	13.4		17	12.6		
	November		14	9.9		18	13.3		
	December		19	13.4		40	29.6		
	January		11	7.7		5	3.7		
	February		13	9.2		6	4.4		
	March		12	8.5		4	3.0		
	April		24	16.9		13	9.6		
	May		13	9.2		2	1.5		
Stadium	Local		72	50.7	0.028 (0.867)	68	50.4	0.007 (0.931)	0.003 (0.956); 0.003
	Visitor		70	49.3		67	49.6		
Time	1st half: 0′–45′		81	57.0	2.817 (0.093)	53	39.3	6.230 (0.013)	8.763 (0.003); 0.175
	0′–15′		29	20.4		6	4.4		
	16′–30′		23	16.2		19	14.1		
	31′–45′ + added time		29	20.4		28	20.7		
	2nd half: 46′–90′		61	43.0		82	60.7		
	46′–60′		26	18.3		20	14.8		
	61′–75′		18	12.7		34	25.2		
	76′–90′ + added time		17	12.0		28	20.7		
Zone	Defensive zone		77	54.2	1.014 (0.314)	69	51.1	0.067 (0.796)	0.269 (0.604); 0.031
	Zone 1		27	19.0		16	11.9		
	Zone 2		21	14.8		20	14.8		
	Zone 3		14	9.9		12	8.9		
	Zone 4		15	10.6		22	16.3		
	Offensive zone		65	45.8		66	48.9		
	Zone 5		16	11.3		13	9.6		
	Zone 6		17	12.0		18	13.3		
	Zone 7		17	12.0		16	11.9		
	Zone 8		15	10.6		18	13.3		
Total accumulated minutes ^c^	0′–500′		30	21.1	19.690 (0.000)	45	33.3	6.600 (0.086)	18.496 (0.000); 0.250
	501′–1000′		30	21.1		33	24.4		
	1001′–1500′		24	16.9		33	24.4		
	>1500′		58	40.8		24	17.8		
Accumulated minutes after resting (>7 days) ^c^	0′–200′		64	45.1	39.014 (0.000)	91	67.4	136.141 (0.000)	17.887 (0.000); 0.246
	201′ –400′		40	28.2		26	19.3		
	401′–600′		21	14.8		5	3.7		
	>600′		17	12.0		13	9.6		
**Cause of injuries**									
Causative agent of the injury	Alone		99	69.7	26.087 (0.000) ^a^	77	57.0	4.038 (0.044) ^a^	4.991 (0.025); 0.135 ^a^
	Opponent		39	27.5		54	40.0		
	Partner		4	2.8		4	3.0		
How the injury is caused	Sprint		41	28.9	118.915 (0.000)	25	18.5	63.667 (0.000)	16.635 (0.164); 0.238
	Dribbling		4	2.8		3	2.2		
	Sliding		4	2.8		4	3.0		
	Turning		11	7.7		5	3.7		
	Tackled by other		13	9.2		21	15.6		
	Shooting		17	12.0		17	12.6		
	Falling		6	4.2		2	1.5		
	Hit by ball		2	1.4		-	-		
	Tackling other player		18	12.7		20	14.8		
	Stretching		6	4.2		5	3.7		
	Collision		10	7.0		16	11.9		
	Kicked by other		3	2.1		9	6.7		
	Unknown		7	4.9		8	5.9		

Note: Coef: contingency coefficient. ^a^: For this analysis, the category with the lowest value for each criterion was discarded. ^b^: This analysis was carried out only for the four most frequent injuries (strain, sprain, contusion and overuse). ^c^: This variable takes into account the accumulated minutes of all competitions played during the season.

**Table 2 ijerph-19-11296-t002:** Analysis of injuries by age and position.

		Spain (LaLiga)				England (Premier League)			
**Age**		**31+**	**26–30**	**21–25**	**20−**	**31+**	**26–30**	**21–25**	**20−**
	Players (*n*)	74	142	107	8	55	166	104	11
	Players (%)	22.4	42.9	32.3	2.4	16.4	49.3	31.0	3.3
	Injured players (*n*)	45	68	28	1	23	70	40	2
	Injured players (%)	31.7	47.9	19.7	0.7	17.0	51.9	29.6	1.5
	Difference (%)	+9.3	+5.0	−12.6	−1.7	+0.6	+2.6	−1.4	−1.8
**Position**		**Gk**	**Def**	**Mid**	**For**	**Gk**	**Def**	**Mid**	**For**
	Players (*n*)	34	134	92	105	26	139	105	92
	Players (%)	9.3	36.7	25.2	28.8	7.2	38.4	29.0	25.4
	Injured players (*n*)	2	76	30	34.0	1	52	41	41
	Injured players (%)	1.4	53.5	21.2	23.9	0.7	38.5	30.4	30.4
	Difference (%)	−7.9	+16.8	−4.0	−4.9	−6.5	+0.1	+1.4	+5.0

Note: Gk: goalkeeper; Def: defender; Mid: midfielder; For: forward.

**Table 3 ijerph-19-11296-t003:** Analysis of injuries by club.

Ranking	Club	Points	Injuries (*n*)	Club	Points	Injuries (*n*)
1	Barcelona	87	9	Manchester City	98	5
2	Atlético de Madrid	76	10	Liverpool	97	8
3	Real Madrid	68	11	Chelsea	72	8
4	Valencia	61	10	Tottenham Hotspur	71	9
5	Getafe	59	4	Arsenal	70	12
6	Sevilla	59	7	Manchester United	66	10
7	Espanyol	53	9	Wolverhampton W.	57	5
8	Athletic Club	53	7	Everton	54	4
9	Real Sociedad	50	14	Leicester City	52	3
10	Real Betis	50	8	West Ham United	52	7
11	Deportivo Alavés	50	6	Watford	50	4
12	Eibar	47	8	Crystal Palace	49	1
13	Leganés	45	4	Newcastle United	45	8
14	Villarreal	44	5	Bournemouth	45	0
15	Levante	44	9	Burnley	40	4
16	Real Valladolid	41	5	Southampton	39	9
17	Celta de Vigo	41	3	Brighton & Hove Albion	36	10
18	Girona	37	6	Cardiff City	34	11
19	Huesca	33	6	Fulham	26	7
20	Rayo Vallecano	32	2	Huddersfield Town	16	3

**Table 4 ijerph-19-11296-t004:** Analysis of the selective search for T-Patterns.

Description of T-Patterns with Five Occurrences	LaLiga	Premier
	N	N
Total T-Patterns detected	12,686	10,818
Non-useful T-Patterns (do not meet selection criterion)	11,655 (91.9%)	10,414 (96.3%)
T-Patterns not excluded	1031 (8.1%)	404 (3.7%)
T-Patterns with strain, sprain, contusion or overuse	4256	2254
T-Patterns with defender, midfielder or forward	3711	2761
T-Patterns with defender and strain	779	104
T-Patterns with defender and sprain	26	98
T-Patterns with defender and contusion	11	32
T-Patterns with defender and overuse	18	0
T-Patterns with midfielder and strain	121	9
T-Patterns with midfielder and sprain	7	6
T-Patterns with midfielder and contusion	0	17
T-Patterns with midfielder and overuse	0	62
T-Patterns with forward and strain	45	58
T-Patterns with forward and sprain	24	13
T-Patterns with forward and contusion	0	2
T-Patterns with forward and overuse	0	3

**Table 5 ijerph-19-11296-t005:** T-Patterns according to league, position (defender, midfielder and forward) and the most frequent injury (strain, sprain, contusion and overuse).

Basic T-Pattern	O	Explanatory T-Pattern	O	L
**LaLiga**				
Defender–strain *	38	((defender (thigh strain))(hamstringfr (alone sprint)))	14	6
		((defender (thigh strain))(hamstringfr alone))	23	5
		* (18) years26_30; (14) more30 years; (21) firsthalf; (18) tam_more1500;		
Defender–sprain *	11	(((days1_19 firsthalf)(defender sprain)) opponent)	5	5
		((defender (knee sprain)) cruciateligrupt)	5	4
		* (8) opponent; (5) tacklingother; (5) tam_more1500; (9) firsthalf; (6) knee		
Defender–contusion	8	(((days20_38 firsthalf)(defender contusion)) tam_more1500)	5	5
		(defender (contusion opponent))	6	3
Defender–overuse	9	((((secondhalf defensivezone)(defender thigh))(rightfooted overuse))(hamstringoveruse alone))	5	8
		((secondhalf (defensivezone defender))(leftleg (overuse alone)))	6	6
		* (7) leftleg; (8) secondhalf; (8) alone; (7) thigh		
Midfielder–strain *	18	(((days1_19 firsthalf)(more30 years midfielder))(thigh (strain alone)))	6	7
		(((firsthalf more30 years)(midfielder thigh)) strain)	7	5
		(((firsthalf midfielder)(strain alone)) amar_0_200)	7	5
		(midfielder (strain (alone sprint)))	9	4
		* (8) more30 years; (16) alone; (12) firsthalf; (11) days1_19; (8) amar_0_200		
Midfielder–sprain *	7	((days1_19 offensivezone)(midfielder sprain))	5	4
		* (5) secondhalf; (5) opponent; (6) local; (6) days1_19		
Midfielder–contusion	0			
Midfielder–overuse	0			
Forward–strain *	13	(offensivezone (((years26_30 forward)(thigh rightfooted))(strain (hamstringfr alone))))	5	8
		(((offensivezone forward)(thigh strain))(hamstringfr (alone amar_0_200)))	7	7
		((forward thigh)(strain hamstringfr))	9	4
		* (8) days1_19; (8) years26_30; (10) offensivezone; (5) october; (10) firsthalf; (7) thigh; (9) hamstringfibrillarruptu; (8) amar_0_200		
Forward–sprain *	10	(days20_38 ((forward ankle)(sprain anklesprain)))	5	5
		* (8) alone; (7) secondhalf; (6) anklesprain		
Forward–contusion	0			
Forward–overuse	0			
**Premier League**				
Defender–strain *	16	((days1_19 defender) (thigh((rightfooted(strainhamstringfibrillarruptu)) (alone amar_0_200))))	10	8
		* (9) secondhalf; (11) 26_30 years; (9) rightleg; (15) alone		
Defender–sprain *	13	((defender leftleg) (sprain opponent))	8	4
		* (10) days1_19; (9) knee; (10) leftleg; (10) opponent;		
Defender–contusion	11	(days1_19 ((defender contusion)(opponent amar_0_200)))	8	5
Defender–overuse	0			
Midfielder–strain *	8	((days20_38 midfielder)((thigh (strain hamstringfibrillarruptu)) alone))	5	6
		* (6) leftleg; (5) 26_30 years; (5) offensivezone; (6) secondhalf; (6) hamstringfibrillarruptu		
Midfielder–sprain *	7	((days1_19 (26_30 years ((midfielder ankle)(rightleg sprain)))) anklesprain)	5	7
		((days1_19 midfielder)(sprain amar_0_200))	6	4
		* (5) opponent; (6) anklesprain; (6) rightleg		
Midfielder–contusion *	10	(secondhalf ((midfielder contusion)(lowerextremitycontusion opponent)))	7	5
		* (10) opponent; (9) secondhalf; (8) lowerextremitycontusion		
Midfielder–overuse *	13	((days1_19 secondhalf)((midfielder overuse)(soleusgastrocnemoveruse (alone amar_0_200))))	6	7
		(secondhalf ((midfielder overuse)(soleusgastrocnemoveruse alone)))	7	5
		((days1_19 secondhalf)((midfielder overuse)(alone amar_0_200)))	7	6
		* (10) secondhalf; (9) days1_19; (7) 21_25 years; (8) amar_0_200		
Forward–strain *	14	(offensivezone (forward ((thigh (strain hamstringfibrillarruptu)) alone)))	10	6
		* (8) days1_19; (5) december; (7) sprint; (5) tam_1001_1500; (12) hamstringfibrillarruptu; (8) secondhalf		
Forward–sprain *	8	(secondhalf ((forward (ankle (sprain anklesprain))) amar_0_200))	6	6
		* (6) visitior; (6) rightleg; (7) secondhalf; (5) days20_38; (5) opponent		
Forward–contusion	6	((offensivezone forward)(contusion opponent))	6	4
Forward–overuse	6	((offensivezone forward)(thigh (overuse alone)))	5	5
		(days1_19 ((offensivezone forward)(overuse alone)))	5	5

Note: TAM: total accumulated minutes; AMAR: accumulated minutes after resting. * Occurrence of the indicated category combined with the basic T-Pattern. O: occurrence; L: length.

## Data Availability

Not applicable.

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
