# Peer review of "Analysis of Injury Patterns in Men’s Football between the English League and the Spanish League"

_ijerph, 2022, doi:10.3390/ijerph191811296_

Round 1

Reviewer 1 Report

This paper analyses the existence of different injury patterns in Spanish and Premier League professional football players. The paper addresses a very interesting and timely topic considering the impact of soccer both from an economic and social point of view. Specifically, the leagues studied represent the two most successful competitions at present. However, the paper has some shortcomings that the authors should tackle to take advantage of the potential of the analysis.

Concerning the introduction, it is stated that “the object of study of the present research is to analyse and determine injury patterns that occurred in LaLiga (Spain) and the Premier League (England) during the 2018/2019 season and to look for differences between these two competitions”

However, beyond presenting the purpose of the research, it would be convenient to add a justification of the practical usefulness of the work and its contribution to knowledge. It would also be useful to add an additional section on practical implications at the end of the conclusions.

One issue that the authors should clarify is the period of analysis, since while in the introduction (line 78) reference is made to the 2018-19 season, a few lines later the Materials and Methods (line 90) refer to the 2017/18 season. The analysis for a single season raises the question of whether this observed behaviour can be extrapolated to other years or whether it is a casual phenomenon at that time.

In relation to the methodology, among others, the authors use the detection of T.Patterns. In this regard, a more detailed explanation of this technique would be desirable so that scholarships less familiar with it can correctly interpret the tables disclosed.

In relation to the results, the authors study the incidence of injuries according to various factors such as the age or player of the footballer. However, in order to be able to interpret the results correctly, it would be necessary to know if there are differences between countries and demarcations in these factors. For example, the results show that, on average, the age at which most injuries occur is 26 to 30 years (47.9% in Spain and 51.9% in the Premier League), which suggests that this is a critical age range. But for that to be true, we would have to know the distribution of percentages by age range in each league, because if the percentage of players between 26 and 30 was equal to or higher than the percentage of injuries, the interpretation would be the opposite. The same should apply to the breakdown by demarcation.
Another element that would be useful to add to the analysis is the distribution of injuries according to club classification, in order to subsequently analyse whether these injuries have a sporting cost for the club or are on the contrary the price to pay to achieve success in the competition.

The discussion requires further analysis in order to better exploit the results and provide useful information for decision making by club managers. In its current version, results, discussion and conclusions are rather repetitive. In some cases, the authors already point to a line that could justify the results, as when they state that the lower number of injuries in players over 30 years of age could be a consequence of the existence of fewer players. This argument should be explored further and evidence should be provided. The discussion should also, in view of the results, provide recommendations to league officials, coaches and club managers to minimise the impact of injuries.

Reviewer 2 Report

The aim of the present study was to analyse and determine injury patterns that occurred in La Liga (Spain) and the Premier League (England) during the 2018/2019 season and to look difference between these competitions. The study is interesting and timely. However, the manuscript needs of improvements, especially in introduction, results and discussion. Please, see comments below:

Introduction:

In general, the introduction is not able to present the importance of study. The authors need to improve the introduction. I have some concerns:

1)  Why the study analyse just La Liga and the Premier League? Based in the literature, the authors need explain this point.

2) Information about intrinsic and extrinsic injury risk factors is approached. However, information about incidence of injuries (type, local, among others) should be considered. In addition, which is the injury incidence in La Liga and the Premier League? Is this incidence higher than other leagues?

3) Why analyse incidence of injuries during competitions? Is the incidence of injuries higher during competition than training?

L34-35: Please cited a reference.

L37-39: Is there information about La Liga?

L42: Which the four major Europe Leagues?

Methods:

L90: Which season was analysed? Introduction claimed that 2018/2019 season was analysed, while methods claimed that 2017/2018 season was analysed.

Why just one season was analysed? Since more three seasons (2019/2020, 2020/2021 and 2021/2022) happened.

L142: Revise citation according to the journal guidelines.

Results

Table 1 and 2: I suggest divide in 4 tables or in 4 topics inside of table 1: 1) Types and local of injuries; 2) intrinsic injury risk factors; 3) extrinsic injury risk factors; and 4) Cause of injuries.

Table 2 did not have title.

In addition, all numbers are presented with “comma”, the numbers should be presented with “dot”

L176: The authors statement “In the Spanish league there were more injuries in the first half (57.0-43.0%)…” No significant association was observed, therefore, this information cannot be claimed.

L177-178: No significant association too.

L187-188: No significant association too.

L188-190: This statement should be first in the results, since is the first information of table.

Table 4: Please show the legend for “O” and “L”.

Discussion

The discussion section remains quite vague and very descriptive. I have some concerns:

1) Why a high number of injuries occur in the defensive zone? Is there relationship with players’ position? Since in the present study defenders were more affected by injuries.

2) As expected, injuries occurred frequently in the lower extremity (L313-316), but the authors may also discuss about type and location of injuries.

3) How the authors explain the most injuries occur without the presence of opponent and after a sprint?

In general, the authors may use more evidence to explain their results.

L295-297: Is there a citation for this statement?

L327-329: Please clarify why the matches during Christmas are associated with injury rate decreasing as players accumulated more minutes in the season.

Thank you.

Reviewer 3 Report

This article shows an analysis of football injuries comparing English and Spanish leagues.

The authors show the importance of this issue and make a deep analysis, doing a good job.

The only thing that i think that should be mentioned is that maybe the authors should take into account not only the accumulated minutes during the season (that affect injuries), but also the accumulated minutes if the players compite with their national football team too, due to this could affect to the total minutes played and increase the probability of injuries.

Even so, i think this article shows very interesting information and i would like to congratulate the authors and see more investigations of this type.

Round 2

Reviewer 1 Report

The work has been substantially improved with the introduction of tables 2 and 3. The added paragraph in the introduction stating the frequency of injuries in professional football is confusing and there seems to be some inconsistency. If the injury incidence rate in the top five European professional leagues is 6.8 injuries/1000 hours of exposure while in professional leagues in other countries it is 7.6 injuries/1000 hours of exposure, the overall average would be between 6.8 and 7.6 but the authors say 8.1 a few lines earlier. In my opinion the whole paragraph could be deleted because it does not add information relevant to the work.

However, tables 2 and 3 add relevant information that the authors should exploit in the discussion and conclusions. The analysis by age and position evidences a different behaviour between League and Premiership that needs to be addressed. Table 2 shows that in La Liga as the age of the player increases the probability of injury increases while in the case of the Premiership age does not seem to be a determining factor. The same table also shows that in La Liga the defence is the position with the highest injury rate while in the Premiership it is the strikers who suffer more injuries than they should. 

On the other hand, the authors should revise the row Difference (%) in the case of age in the Spanish league, as the total sum should be 0 and it is not (the higher injury rate for a certain age should be compensated by a lower one for another age).

Table 3 also seems to indicate that excessive injuries in clubs do not have an impact on their sporting performance. However, the way the information is presented is a bit strange because while in the LR column the lower values indicate better results (1 is the champion and 20 is the last), in the case of the IR column the arrangement is the other way around (1 is the team with the most injuries and 20 the one with the least injuries). I would suggest that students use a correlation analysis between the number of injuries and the points obtained in the ranking and study the sign and significance of the relationship.

Reviewer 2 Report

The authors improved significantly the manuscript. Thank you for considering the comments. I have a minor concern:

L224 and L228: Please change “Table 3” for “Table 4”.

L230 and L232: Please change “Table 4” for “Table 5”.
